# A Humanized and Viable Animal Model for Congenital Adrenal Hyperplasia–*CYP21A2*-R484Q Mutant Mouse

**DOI:** 10.3390/ijms25105062

**Published:** 2024-05-07

**Authors:** Shamini Ramkumar Thirumalasetty, Tina Schubert, Ronald Naumann, Ilka Reichardt, Marie-Luise Rohm, Dana Landgraf, Florian Gembardt, Mirko Peitzsch, Michaela F. Hartmann, Mihail Sarov, Stefan A. Wudy, Nicole Reisch, Angela Huebner, Katrin Koehler

**Affiliations:** 1Division of Paediatric Endocrinology and Diabetes, Department of Paediatrics, Faculty of Medicine and University Hospital Carl Gustav Carus, Technische Universität Dresden, 01307 Dresden, Germany; shaminiramkumar.thirumalasetty@ukdd.de (S.R.T.); tinaschubert05@gmail.com (T.S.); marie-luise.rohm@tu-dresden.de (M.-L.R.); dana.landgraf@ukdd.de (D.L.); angela.huebner@ukdd.de (A.H.); 2Transgenic Core Facility, Max Planck Institute of Molecular Cell Biology and Genetics, 01307 Dresden, Germany; naumann@mpi-cbg.de; 3Genome Engineering Facility, Max Planck Institute of Molecular Cell Biology and Genetics, 01307 Dresden, Germany; reichard@mpi-cbg.de (I.R.); sarov@mpi-cbg.de (M.S.); 4Division of Nephrology, Medizinische Klinik III, Faculty of Medicine and University Hospital Carl Gustav Carus, Technische Universität Dresden, 01307 Dresden, Germany; florian.gembardt@ukdd.de; 5Institute of Clinical Chemistry and Laboratory Medicine, Faculty of Medicine and University Hospital Carl Gustav Carus, Technische Universität Dresden, 01307 Dresden, Germany; mirko.peitzsch@ukdd.de; 6Steroid Research & Mass Spectrometry Unit, Paediatric Endocrinology and Diabetology, Center of Child and Adolescent Medicine, Justus Liebig Universität, 35392 Giessen, Germany; michaela.hartmann@paediat.med.uni-giessen.de (M.F.H.); stefan.wudy@paediat.med.uni-giessen.de (S.A.W.); 7Medizinische Klinik und Poliklinik IV, LMU Klinikum München, 80336 Munich, Germany; nicole.reisch@med.uni-muenchen.de

**Keywords:** adrenals, CAH, animal models

## Abstract

Congenital Adrenal Hyperplasia (CAH) is an autosomal recessive disorder impairing cortisol synthesis due to reduced enzymatic activity. This leads to persistent adrenocortical overstimulation and the accumulation of precursors before the blocked enzymatic step. The predominant form of CAH arises from mutations in *CYP21A2*, causing 21-hydroxylase deficiency (21-OHD). Despite emerging treatment options for CAH, it is not always possible to physiologically replace cortisol levels and counteract hyperandrogenism. Moreover, there is a notable absence of an effective in vivo model for pre-clinical testing. In this work, we developed an animal model for CAH with the clinically relevant point mutation p.R484Q in the previously humanized *CYP21A2* mouse strain. Mutant mice showed hyperplastic adrenals and exhibited reduced levels of corticosterone and 11-deoxycorticosterone and an increase in progesterone. Female mutants presented with higher aldosterone concentrations, but blood pressure remained similar between wildtype and mutant mice in both sexes. Male mutant mice have normal fertility with a typical testicular appearance, whereas female mutants are infertile, exhibit an abnormal ovarian structure, and remain in a consistent diestrus phase. Conclusively, we show that the animal model has the potential to contribute to testing new treatment options and to prevent comorbidities that result from hormone-related derangements and treatment-related side effects in CAH patients.

## 1. Introduction

Congenital adrenal hyperplasia (CAH) is a disorder of adrenal steroidogenesis, classified as one of the most frequent recessive inherited disorders. It is primarily caused by 21-hydroxylase deficiency (21-OHD) resulting from *CYP21A2* mutations. There are two forms of CAH: classic CAH, which is severe and life-threatening, attributable to the partial and complete ablation of residual enzyme activity and resulting in a lack of cortisol, and non-classic CAH, which is mild and caused by *CYP21A2* mutations that retain almost 20–50% residual enzyme activity and sustained cortisol biosynthesis [1]. While classic and non-classic CAH differ in age of onset, symptoms, and severity, all patients primarily manifest with hyperandrogenism [2]. Homozygous and compound heterozygous point mutations are typical for 21-OHD. In recent years, over 100 mutations in the *CYP21A2* gene have been identified to be associated with either the simple virilizing or the severe salt-wasting phenotypes of the classic form, as well as the milder non-classic phenotypes [3,4]. An example of such a mutation is the p.R484Q (Arg484Gln), resulting from a DNA alteration at position c.1451G > A in the exon 10 of the *CYP21A2* gene [5]. The missense mutation p. R484Q resides in the C-terminal end of the 21-hydroxylase (21-OH) protein. In vitro assays with the point mutation R484Q revealed that there is a low but measurable activity of 21-OH-R484Q for the substrate conversion of 17-hydroxyprogesterone (1.1%; SD 0.7%) and progesterone (3.8%; SD 1.9%), the two natural substrates of 21-OH [6]. Previous research on *CYP21* mutations has indicated that in vitro activities between 1–14% correlate with the simple virilizing phenotype, thereby suggesting that the R484Q missense mutation is part of *CYP21* mutations that are present in the simple virilizing form of the disorder [7,8,9].

Individuals diagnosed with CAH require continuous medical intervention to compensate for deficient hormones and to regulate elevated androgen levels. Previous studies have shown that complications have persisted due to exposure to high doses of glucocorticoids and the continued high level of adrenal androgens [10]. Over the last few years, there has been substantial improvement in the development of new drugs and therapies for CAH. Concomitantly, the testing of new therapies poses a challenge as the existing models have limited efficacy and are quite restrictive in scope for the disorder [11]. 

The first genetically modified mouse model for 21-hydroxylase deficiency was the C57BL/10SnSlc-H-2^aw18^ knockout mouse strain that showed increased levels of progesterone, typical to CAH. These H-2^aw18^ mice were, however, difficult to handle and showed early postnatal mortality without treatment [12]. Survival was therefore ensured by rigorous dexamethasone substitution for both dams and pups [13,14]. National Center of Biotechnology Information (NCBI) blast analysis shows that there is a 73% amino acid and 78% nucleotide sequence homology between the human *CYP21A2* and mouse *Cyp21a1* genes. Therefore, to study human mutations and further delineate CAH, we developed the humanized mouse model. In our previous work, we have shown that the humanized C57BL/6NCrl-*Cyp21a1*^tg(*CYP21A2*)koe^ mice showed no differences in fertility, viability, and growth and, therefore, have the potential to be an excellent model to introduce mutations relevant to CAH [11,15].

In this study, we generated and characterized a novel humanized CAH mouse model in which we introduced the CAH-causing mutation p.R484Q into the humanized C57BL/6NCrl-*Cyp21a1*^tg(*CYP21A2*)koe^ mice. The *CYP21A2*-R484Q mice show typical CAH characteristics with hyperplastic adrenals, low levels of corticosterone, and higher concentrations of progesterone. This mouse model is the first viable rodent model available to test treatment and therapies relevant to CAH and has the potential to facilitate the transition from basic research into clinical application. 

## 2. Results

### 2.1. Development of a Humanized CYP21A2-R484Q Knock-in Mouse Model

Earlier studies have developed models related to CAH; however, these models faced challenges due to restrictive survival rates and were potentially less comprehensive in addressing the full scope of the disorder. Previously, we successfully characterized the humanized C57BL/6NCrl-*Cyp21a1*^tg(*CYP21A2*)koe^ mice [15]. In this work, we demonstrate the impact of the integration of the human pathogenic variant p.R484Q into the *CYP21A2* gene of the humanized knock-in mouse. Our approach includes the design of the knock-in DNA fragment that targets the desired genetic modification via CRISPR/Cas, resulting in the amino acid exchange of arginine 484 to glutamine (p. Arg484Gln; R484Q) in zygotes derived from the humanized *CYP21A2* strain. PCR and Sanger sequencing were used to confirm the presence of the knock-in allele in the resulting pups, as shown in Appendix A. To produce a population of h*CYP21A2*-R484Q, a mutation-carrying female mouse with the correct integration was mated with C57BL/6NCrl wildtype (WT) male. Subsequently, heterozygous offspring were intercrossed to yield homozygous (HOM) mutated mice. The developed mouse strain is designated according to the international nomenclature as C57BL/6NCrl-*Cyp21a1*^tg(*CYP21A2*)koe-R484Q^. In this paper, the terms mutant mice or HOM are used to refer to the transgenic mouse line, WT is used for the wildtypes, and HET is used to refer to the heterozygous mice. 

The genotype frequencies in the offspring have a slight deviation from the Mendelian genetics with a distribution of 1:2:0.6 (WT:HET:HOM). Dividing by sex, this distribution shifted to 1:2:0.7 for males and 1:2:0.5 for females. The chi-square test was utilized to evaluate the correspondence between the observed distribution and the expected Mendelian distribution. The obtained chi-square statistics for the male mice imply that the observed distribution did not significantly differ from the expected Mendelian distribution (*p* = 0.3752). In contrast, in the case of the female mice, there was an indication of a potential deviation from the Mendelian distribution (*p* = 0.056). No differences in viability and growth were seen between mutant and WT mice. The pups showed no abnormalities in a severity assessment in the first week after birth. There was a normal litter size with an average of 6.8 pups per litter (1–11 pups; SD ± 2.9) with a mean loss of 0.6 ± 1.4 pups per litter until the day of weaning (day 21). The sex distribution was slightly shifted towards males with 54.27% male and 45.73% female. Body weights of 20-week-old male mutant mice showed no significant differences in comparison to their wildtype littermates (33.63 g ± 0.93 g vs. 32.33 g ± 0.46 g; *p* = 0.1699). The body weight of the mutant female mice was significantly lower than their WT littermates (24.57 g ± 0.63 g vs. 28.59 g ± 0.98 g; *p* = 0.006), as depicted in Figure 1A. As expected, due to adrenal hyperplasia, the adrenals of the mutant mice weighed more than their wildtypes (0.0059 g ± 0.0003 vs. 0.0032 g ± 0.0003 g, *p* = 0.0003; 0.0078 g ± 0.0012 g vs. 0.0039 g ± 0.0016 g, *p* = 0.0007), as shown in Figure 1B.

### 2.2. Steroidogenic Enzyme Expression Levels 

The expression levels of steroidogenic enzymes were studied to assess the gene expression profile of the mutant mice in comparison to their wildtype (WT) littermates.

As expected, adrenals showed a complete ablation of the expression of gene *Cyp21a1* in both male and female mutant mice (*p* < 0.0001; *p* = 0.003) (Figure 2A), while the expression of gene *CYP21A2* was activated in equimolar amounts (*p* < 0.0001; *p* = 0.0003) (Figure 2B). *Cyp11a1* expression levels between male animals did not differ; however, there was a significant difference in *Cyp11a1* expression between female wildtype and mutant animals (*p* = 0.0059) (Figure 2E). *Cyp11b1*, *StAR*, *Hsd3b2*, and *Nr5a1* expression levels did not differ in either male or female animals between the genotypes (Figure 2C,F–H). In addition, the mutant animals showed a significantly higher expression of the *Cyp11b2* gene (*p* = 0.0001; *p* = 0.0003) (Figure 2D). In conclusion, mutant animals are characterized by a complete loss of *Cyp21a1* expression with a replaced expression of *CYP21A2* and upregulated *Cyp11b2* expression. In addition, female mutants exhibit an increased *Cyp11a1* expression.

The gonads were also examined for steroidogenic expression levels. In the ovaries, *Cyp11a1* (*p* = 0.028) and *StAR* (*p* = 0.0007) expression were markedly decreased, whereas *Hsd3b2* expression levels were unchanged (Figure 3A–C). In contrast, expression levels of *Cyp11a1*, *StAR*, and *Hsd3b2* in testes did not differ between WT and mutants (Figure 3D–F). In comparison to the adrenals, all the investigated genes were expressed in low concentrations in ovaries and testes. The expression levels of pro-opiomelanocortin (POMC), an essential precursor protein of the production of adrenocorticotrophic hormone (ACTH), were measured in the pituitary gland of 20-week-old mice (Figure 3G). Surprisingly, no significant differences were measured between mutant animals and their wildtype littermates, thereby contradicting the hyperplasia of the adrenals. ACTH was measured using the plasma of 20-week-old mice (Figure 3H). Significantly higher ACTH levels were measured in the HOM male mutants in comparison to their wildtype littermates (*p* = 0.0008). Interestingly, no differences were noted among the females (*p* = 0.5033). 

### 2.3. Steroidogenic Hormone Concentration Levels 

Heparinized blood plasma collected from eight- and 20-week-old mice was used to analyze the steroid hormone levels. At eight weeks, we measured significant differences in progesterone (*p* = 0.001) and corticosterone (*p* = 0.003) levels between the male mutant mice and their wildtypes. Although a tendency to change was visible, no significant hormone differences were noted for other measured hormones (Appendix A). The limit of detection and limit of quantification values measured during steroid hormone measurements using LC-MS/MS is described in Appendix A. 

At 20 weeks, progesterone levels were markedly elevated in both male and female mutant animals (54.85 ng/mL ± 10.06 ng/mL, *p* < 0.0001; 74.83 ng/mL ± 21.25 ng/mL, *p* < 0.0001) in comparison to the respective wildtype littermates (2.57 ng/mL ± 1.89 ng/mL; 2.51 ng/mL ± 0.58 ng/mL) (Figure 4A). Comparing both sexes, female mutants showed higher concentrations of progesterone than males, hinting that the sex hormones can play a part in this modulation. The 11-deoxycorticosterone levels in the mutant animals (0.32 ng/mL ± 0.05 ng/mL; 0.61 ng/mL ± 0.12 ng/mL) were significantly lower in comparison to wildtype animals (*p* < 0.0001; *p* = 0.0025) (Figure 4B). Furthermore, these reduced levels were further reflected by lower concentrations of corticosterone. Corticosterone levels were significantly reduced in both male and female mutant mice in comparison to their wildtype littermates (42.54 ng/mL ± 6.52 ng/mL vs. 119.90 ng/mL ± 15.15 ng/mL, *p* = 0.0003; 64.97 ng/mL ± 7.89 ng/mL vs. 185.50 ng/mL ± 25.91 ng/mL, *p* = 0.0009) (Figure 4C). 18-hydroxycorticosterone levels did not significantly differ between the mutants and their wildtype controls (Figure 4D). Female mutant mice showed significantly increased concentrations of aldosterone in comparison to their wildtype littermates (0.63 ng/mL ± 0.08 ng/mL vs. 0.31 ng/mL ± 0.04 ng/mL, *p* = 0.0059) and were higher than in the male mutants (0.37 ng/mL ± 0.03 ng/mL) (Figure 4E). Testosterone levels did not differ between the male mutants and their wildtype controls and remained at negligible levels in females (Figure 4F). 

### 2.4. Blood Pressure Measurements

Assessing blood pressure is important for the evaluation of overall health and potential hormonal influences. The mice were subjected to a non-invasive and reliable method of measurement using a tail-cuff. At eight weeks, significant differences were measured between the mutant females and their wildtypes for mean arterial pressure (MAP) (*p* = 0.026), systolic blood pressure (SBP) (*p* = 0.042), and diastolic blood pressure (DBP) (*p* = 0.014) (Appendix A). 

The MAP showed no notable distinctions between mutants and wildtypes in either sex (male: 40.8 mmHg ± 9.62 mmHg, *p* = 0.459; female: 45.38 mmHg ± 6.57 mmHg, *p* = 0.866) (Figure 5A). Similarly, no differences were noticed in the systolic (*p* = 0.530; *p* = 0.463) or diastolic blood pressures (*p* = 0.530; *p* = 0.778) or the heartrate (*p* = 0.6389; *p* = 0.152) between the wildtypes and mutants (Figure 5B–D).

### 2.5. Measurement of Urine Metabolites

A tetrahydrogenated progesterone metabolite with three additional hydroxyl groups was detected in high concentrations in the urine of mutant animals (1769 µg/L ± 274.6 µg/L, *p* < 0.0001; 1181 µg/L ± 244.1 µg/L, *p* < 0.0001), presenting an excellent biomarker of the enzyme deficiency (Figure 6A). Another significant urine metabolite, a tetrahydrated metabolite of the dominant mouse glucocorticoid corticosterone (tetrahydrocorticosterone), showed higher concentrations in female wild types than in female mutants (590.7 µg/L ± 131.8 µg/L, *p* = 0.001), whereas in male mice no tetrahydrocorticosterone was detectable in either wildtypes or mutants (Figure 6B). 

### 2.6. Organ Morphology and Histology

Histological examinations were conducted using tissue samples from 20-week-old mice to detect any anomalies in the morphology and cellular structure of a few important mouse organs (brain, liver, kidneys, adrenal glands, ovaries, and testes). Macroscopic imaging indicated an enlargement of the adrenal size of mutant animals in comparison to their respective wildtype littermates (Figure 7A). The morphology and zonation of glands were examined by H&E staining (Figure 7B–G). Following the higher adrenal weight, a prominent hyperplasia in the zona fasciculata was evident in the adrenal sections of both male and female mutant mice (Figure 7C,E). The adrenal medulla seemed macroscopically unchanged. Immunofluorescence studies conducted using a medulla-specific marker tyrosine hydroxylase (TH) on adrenal cryosections depicted the formation of a normal medulla in the mutant mice and confirmed a sharp distinction between the cortex and the medulla (Figure 8A–D). The testes did not show notable histological alterations (Figure 7H,I). Male mutant animals developed normally and were fertile. In contrast, the ovaries of mutant female mice differed from their wildtype littermates. There was a marked reduction in the numbers of the corpus lutea (Figure 7J,K). Furthermore, the ovaries of mutant females also depicted a regression of follicles, indicating hindrances in follicular development (Figure 7K). Histological investigations of the brain, liver, and kidney revealed no anomalies in morphology or cellular structure.

## 3. Discussion

In this study, we generated and characterized a knock-in mouse model integrated with a CAH pathogenic variant, p.Arg484Gln, by the name R484Q in the humanized mouse line C57BL/6NCrl-*Cyp21a1*^tg(*CYP21A2*)koe^. CAH triggered by 21-OH deficiency is considered a fully penetrant genetic condition showing a strong, although not complete, genotype–phenotype relationship in which the less severely affected allele determines clinical symptoms. 

Most of the patients with 21-OHD present as compound heterozygotes for various mutations. Patients exhibit either complete deletions or crucial gene conversions affecting the entire *CYP21A2* gene. Furthermore, the nine most frequently occurring point mutations are also present in pseudogene *CYP21A1P*, sharing 98% similarity with the coding gene sequence [6,16]. The genetic similarity and proximity between pseudogene *CYP21A1P* and functional gene *CYP21A2* facilitate various cross-overs accounting for over 90% of all documented mutations linked to 21-hydroxylase deficiency [17]. A comparative description of the differences between the human CAH patients and the *CYP21A2*-R484Q mouse model is provided in Table 1.

As in humans, the murine genome also has both the functional gene *Cyp21a1* and a corresponding pseudogene *Cyp21a2-p*. The murine gene is positioned at chromosome 17, near the mouse H-2 class III region of the major histocompatibility complex (MHC) locus, and encodes a protein composed of 487 amino acids. The murine and human genes show a genetic homology of more than 75%, whereas their respective protein products share a 72.5% similarity [15]. Although CAH is one of the most common disorders of adrenal steroidogenesis, advances in developing a suitable mouse model to study new treatments and therapies have been quite limited. Our knock-in mouse model with the point mutation p.R484Q in h*CYP21A2* on a C57BL/6NCrl genetic background exhibits an impaired steroid metabolism in the mutant mice. We suspect that the genotypic frequencies of the mice slightly deviated from the expected Mendelian distribution probably due to the loss of pups triggered by the phenotypic conditions of the mutant mice. Gene expression analysis indicates that the h*CYP21A2* gene under the control of the mouse promoter can fully substitute for the expression of the m*Cyp21a1* gene. While cortisol is the major endogenous adrenal steroid in humans, it is corticosterone for mice, which has the highest blood plasma concentration in comparison to other steroids produced by the adrenal gland [18]. 11-deoxycorticosterone and corticosterone were measurable in significantly lower concentrations in the blood plasma of both male and female mutant mice in comparison to their wildtype littermates, indicating 21-hydroxylase deficiency (21-OHD). Progesterone, the corticosteroid precursor, was significantly elevated in the plasma of mutant animals in comparison to their respective wildtypes. Its counterpart in humans, 17-hydroxyprogesterone (17-OHP), serves as a successor metabolite to progesterone; notably, elevated levels of 17-OHP are identified as a primary diagnostic indicator for CAH in human patients. The accumulation of precursor molecules in the blood of mutant mice implies an enzymatic impairment of the subsequent steps in adrenal steroidogenesis. In 21-OH deficiency, there is an accumulation of both progesterone and 17-OHP due to the impairment in the conversion process. The histological analysis of mouse tissues shows a marked hyperplasia of the adrenal cortex of the mutated animals, indicating poor corticosterone production. Inefficient cortisol production in human CAH patients triggers the production of elevated levels of ACTH, thereby causing the hyperplasia of adrenocortical steroid-producing cells and leading to the development of hyperplasia [19]. Although we observed higher ACTH concentrations in the male mutant mice, the same could not be seen in the female mutants. This is in contrast to the marked differences in ACTH levels seen in 21-OH knock-out mice [20]. In addition, we did not succeed in observing significantly elevated expression levels of POMC. We suspect this might be a methodological issue or be due to the nature of the mutation and the residual enzyme activity. However, based on the observation of adrenocortical hyperplasia seen in mutant mice, we deduce that ACTH must be presumably elevated in the female mice as well, analogous to human CAH patients.

Aldosterone, another major glucocorticoid, is fundamentally responsible for maintaining salt and water homoeostasis and blood pressure regulation [21]. Unlike male mice, the female mutant mice have significantly higher concentrations of aldosterone than their corresponding wildtypes. Although aldosterone is the primary regulator of blood pressure and blood volume, blood pressure did not differ between the mutant mice and their wildtypes for both sexes. Therefore, sexual dimorphism was only observed in aldosterone levels. However, the underlying mechanism for this disparity remains uncertain.

We were also successful in determining steroid hormone biomarkers of 21-OHD in mouse urine. Applying the highly specific technique of GC-MS urinary steroid analysis, we identified a metabolite of progesterone with a saturated ring system and three additional OH-groups. The fragmentation pattern suggested one of these groups to be located at position 16. The other two OH-groups are probably in position 11 and position 6, the latter being a typical hydroxylation site in the mouse. This parameter enabled a strong differentiation between the mice that were affected and those that were unaffected, aligning with the observations regarding progesterone levels in the plasma. The other metabolite is tetrahydrogenated (ring-reduced) corticosterone, the main glucocorticoid in mice. Taken together, these findings reflect the availability of a non-invasive monitoring of future treatment approaches.

Cyp11a1, a cholesterol side-chain cleavage enzyme, is responsible for catalyzing the first reaction in the steroidogenic pathway, the conversion of cholesterol to pregnenolone. *Cyp11a1* gene expression was significantly higher in female mutant mice in comparison to their wildtype littermates but not in male mice. In contrast, the expression of steroidogenic acute regulatory protein (*StAR*), responsible for regulating the transfer of cholesterol from the outer to the inner mitochondrial membrane, did not differ between mutant animals and their wildtype littermates. The gene expression levels of *Cyp11b1* did not change between mutant and wildtype groups, which rather contradicts the reduced corticosterone levels in mutant animals. Interestingly, the expression of *Cyp11b2*, an enzyme responsible for producing aldosterone, was significantly higher in both male and female mutant animals in comparison to their wildtype littermates. This is inconsistent with our finding of elevated aldosterone levels in the female mutant mice. 3ß-hydroxysteroid dehydrogenase (Hsd3b2), an enzyme catalyzing the conversion of pregnenolone to progesterone, did not vary between mutant and wildtype animals in both sexes. 

Under normal conditions, the follicular and luteal phases of the menstrual cycle play a crucial role in the release of progesterone. However, untreated CAH leads to continuous adrenal hypersecretion and provokes a wide range of contraceptive effects. Furthermore, persistently high progesterone levels might impair folliculogenesis, resulting in follicular regression subsequently impairing follicle implantation [22,23]. This could be one explanation for our observations in the female R484Q mutant mice, where we noticed fewer follicles and signs of follicular regression. In mice, the estrus cycle is staged in four phases, proestrus, estrus, metestrus, and diestrus, repeating every 4–5 days. The diestrus phase in mice is found to be homologous to the human late secretory phase and is associated with high progesterone levels [24]. When assessed for the estrus cycle over seven days, the female mutant mice were found to be in a consistent diestrus phase and showed signs of follicular regression. Therefore, female mutant mice projected fertility disturbances and were sterile. This condition of fertility impairment is comparable to the profound impact of CAH (21-OHD) on the ability of patients to reproduce [23]. In the future, we intend to delve deeper into the fertility of the mutant mice and draw analogies to human patients. The testes of the mutated mice did not show any histological differences and had no fertility disturbances, unlike the female mutant mice. 

Some other notable animal models in the field include the C57BL/10SnSlc-H-2^aw18^ mouse strain, which was difficult to keep alive without the constant supply of dexamethasone to both the dams and pups [12,25]. Another example includes a *StAR* knockout model developed to understand congenital lipoid adrenal hyperplasia; a condition characterized by impaired steroidogenesis with typical lipid deposits in the steroidogenic tissues. The mouse strain was generated by disrupting the *StAR* gene by deleting a part of exon 2 and all of exon 3. Corticosteroids and 0.9% sodium chloride were supplied to rescue the knockout mice during their lifetime [11,26]. A more recent model for CAH includes the *Cyp11b1* null mouse, which was generated by exchanging exons 3–7 of *Cyp11b1* with the cDNA encoding for the cyan fluorescent protein. The mice were viable but suffered from mineralocorticoid excess, hypertension, glucose intolerance, and female infertility [27]. 

In recent years, mice carrying a functional human gene have gained popularity in the field of biomedical research. Model animals are inherently different from humans in various aspects, but humanized mice offer the opportunity to study the development and functionality of human genes in vivo. Despite the various limitations associated with humanized mice and a need for caution while interpreting the obtained results, there is reasonable proof that their use is instrumental in the delineation of biomedical research [28]. In general, the generated knock-in mouse model can be considered a significant step in exploring the functional significance of the *CYP21A2* gene. The integration of the point mutation c.1451G>A p.Arg484Gln has led to the development of a viable mouse line which can be beneficial in testing new treatments and therapies associated with CAH. The mutation p.R484Q is associated with the simple virilizing form characterized by the 21-OH enzymatic activity ranging from 1% to 4% [6,8]. One of the limitations associated with this CAH mutant mouse is that they will not be able to reproduce the excessive secretion of androgens observed in CAH patients. Unlike humans, the mouse adrenal cortex cannot produce androgens (specifically DHEA) due to the absence of *Cyp17a1* expression in the adrenal, as it is restricted to the gonads only. Overcoming the lack of adrenal *Cyp17a1* expression will be one focus of our future research. 

In summary, the established transgenic mouse model integrated with the point mutation p.R484Q will be beneficial as an in vivo testing system for developing treatments and therapies for CAH. The thorough characterization of the humanized CAH mouse model has demonstrated that the human 21-hydroxylase enzyme effectively replaces the function of the mouse 21-hydroxylase enzyme. Mutant homozygous mice exhibit elevated progesterone levels with reduced 11-deoxycorticosterone and corticosterone levels, measurable in serum and urine. The adrenocortex of both male and female mutant mice were markedly hyperplastic. The 21-OHD led to the production of elevated progesterone, thereby influencing the development of follicles and influencing estrous cycle (Appendix A). 

In conclusion, humanized homozygous mutant *CYP21A2* mice not only serve as a robust model for studying the 21-hydroxylase function and the metabolic aspects of CAH, but they also represent a valuable model for clinical applications such as the testing of novel therapies in the future.

## 4. Materials and Methods

### 4.1. Sex as a Biological Variable

In our study, we examined male and female animals, and sex-dimorphic effects are reported if any. 

### 4.2. Experimental Animals

All animal experiments were performed in compliance with the approved standards of animal care as described in the Ethical Guidelines of the German Animal Welfare Act and Directive 2010763/EU, which protects all animals used for scientific purposes. C57BL/6NCrl mice were purchased from Charles River (Sulzfeld, Germany). *CYP21A2*-R484Q heterozygous mice were generated and bred on a C57BL/6NCrl genetic background. The breeding and maintenance of the mouse strain were accomplished at the mouse facility of the Centre of Regenerative Therapies Dresden (CRTD), Technische Universität Dresden, Germany. The mouse facility was maintained under specific pathogen-free conditions in individually ventilated cages on a 12 h day/night cycle. Standard mouse chow and autoclaved sterile water ad libitum were supplied to the mice.

### 4.3. Integration of the Point Mutation Arg484Gln (p.R484Q) into Humanized CYP21A2 Mice

Humanized *CYP21A2* mice were generated previously by replacing the 2620 bp *Cyp21a1* mouse gDNA sequence with its orthologous 2713 bp *CYP21A2* human gDNA sequence using CRISPR/Cas9-mediated gene targeting [15]. *CYP21A2*-R484Q mice were generated using the CRISPR/Cas9 technique with a specific single-guide RNA and a homology-direct repair oligo in zygotes obtained by intercrossing wildtype females with humanized *CYP21A2* males. Guide RNAs close to the specific point mutation site in h*CYP21A2* gene were chosen by low off-target activity and assessed using http://crispor.tefor.net (accessed on 2 April 2020). The guide RNA 5′-CAAGTGCGGCTGCAGCCCCG-3′ was obtained as crispr RNA (crRNA) (Integrated DNA Technologies, Inc. (IDT), Coralville, IA, USA). The homology-directed repair oligo 5′-TCATCCTCAAGATGCAGCCTTTCCAAGTGCGGCTCCAACTCAGGCATGGGCGCTCACAGCCCGGGCCAGAGCCAGTGACAGGATAGGA-3′ was designed to contain the desired point mutation p.R484Q in addition to several silent mutations to prevent the re-cutting of the repaired allele and to facilitate genotyping. The repair oligo was ordered as custom PAGE purified Ultramer^TM^ DNA oligos (IDT, Coralville, IA, USA). Electroporation was used to integrate the point mutation into the generated zygotes and was performed using a 1 mm cuvette (BioRad, 1652089) and a BioRad Gene pulser XCell electroporator (Bio-Rad Laboratories GmbH, Feldkirchen, Germany). To electroporate the zygotes, Cas9 ribonucleoprotein particles were assembled by combining 4 µM Alt-R^®^ S.p. HiFi Cas9 nuclease V3 (10 mg/mL) (IDT, Leuven, Belgium) with 4 µM of crRNA:tracrRNA duplex and 10 µM of single-strand oligodeoxynucleotide (ssODN) in 10 µL Opti-MEM medium (Gibco^TM^ 31985062; Thermo Fisher Scientific, Darmstadt, Germany). Around 4 h post-electroporation of the CRISPR/Cas9 mix into fertilized zygotes, the surviving embryos were transferred into pseudo-pregnant recipient female mice. Positive pups carrying the point mutation p.R484Q were screened using PCR with primer pairs 5′-GCTCCAACCTCAGGGCAT-3′ and 5′-CTTCCCTTGACAACCCTCTCC-3′, which specifically amplify modified alleles. Furthermore, Sanger Sequencing was used to verify positive candidates using primer pairs spanning the mutated site. Founder *CYP21A2*-R484Q heterozygous mice were bred on a C57BL/6NCrl genetic background. Next, two heterozygous mice with the point mutation were intercrossed to generate the homozygous mice.

### 4.4. Genotyping

Genomic DNA was isolated from the ear biopsies of the three-week-old transgenic mice using the One Step Mouse Genotyping Kit (Vazyme Internationa LLC, San Diego, CA, USA) in accordance with the manufacturer’s protocol. Two PCR reactions were performed to differentiate between the wildtype and the *CYP21A2* allele and to verify the presence of the R484Q mutation. For genotyping, two forward primers, 5′-GTGTCATCCTCAAGATGCAG-3‘ and 5‘-CCAAGACCAGGGTGAGCGT-3‘, and one reverse primer, 5′-CTCACACCCCAGTAGAGAAG-3′ (Eurofins, Ebersberg, Germany), resulted in a 165 bp fragment for the wildtype mice, a 222 bp for the homozygous mice, and a double band for heterozygotes containing fragments of both wildtype and homozygous. Further on, a mutation-specific PCR reaction was performed using the primers 5′-GCTCCAACCTCAGGGCAT-3′ and 3′- CTTCCCTTGACAACCCTCTCC-5′ (Eurofins, Ebersberg, Germany). This resulted in a 322 bp fragment for HOM and HET mice and no fragment for the wildtypes. To further validate the results, Sanger sequencing was performed using the primers 5′-CTGCCGTGAAAATGTGGTGG-3′ and 5′-CTTCCCTTGACAACCCTCTCC-3′.

### 4.5. Mouse Specimen Collection

Blood samples were collected from eight-week-old mice in heparin tubes from the retrobulbar venous plexus. The plasma recovered from the blood samples was stored at −80 °C and subjected to only one freeze-thaw cycle. Between 20–22 weeks, the mice were anaesthetized with a ketamine–xylazine solution and subjected to a final cardiac puncture to collect blood, after which they were sacrificed by cervical dislocation. The brain, pituitary, liver, kidneys, adrenal glands, ovaries, and testes were removed and were both macroscopically and microscopically investigated. For microscopic investigations, the organs were stored at 4 °C and later processed for histological staining. For gene expression analysis, the organs were snap-frozen in liquid nitrogen and preserved at −80 °C for analysis via quantitative PCR. Metabolic cages were used to collect urine for a 24-h period, during which food and water consumption as well as weight were monitored. The urine was further stored at −80 °C and analyzed for urine metabolites using gas chromatography-mass spectrometry (GC-MS). 

### 4.6. Total RNA Extraction and Quantitative Reverse Transcription PCR

The adrenals, pituitaries, ovaries, and testes of 20-week-old mice preserved at −80 °C were used to extract total RNA using NucleoSpin RNA II kit (Macherey-Nagel, Düren, Germany) according to the manufacturer’s protocol. The purity of the RNA was assessed using the NanoDrop Spectrophotometer (ND-1000) (NanoDrop Technologies, Wilmington, DE, USA). One µg of total RNA was reversely transcribed into cDNA using the GoScript Reverse Transcription System (Promega, Mannheim, Germany) according to the manufacturer’s protocol. Specific gene amplification primers for *CYP21A2*, *Cyp11b1*, *Cyp11b2*, *Cyp11a1*, *StAR, Hsd3b2*, and *Nr5A1* were designed using Primer Express 3.0 and sourced from Eurofins Genomics (Ebersberg, Germany). For the analysis of *Cyp21a1* and POMC expression, a TaqMan Gene Expression Assay (Thermo Fisher Scientific, Darmstadt, Germany) was purchased. Real-time PCR was conducted using the GoTaq^®^ Probe qPCR MasterMix (Promega, Mannheim, Germany) in accordance with the manufacturer’s guidelines. Each sample was analysed in triplicate using the Quantstudio 5 (Thermo Fisher Scientific, Darmstadt, Germany. The genes were normalized using the housekeeping gene β-Actin (Actb) as the reference gene. The delta-delta threshold cycle (ΔΔCt) method was utilized to calculate the changes in mRNA expression levels [29]. 

### 4.7. Plasma Steroid and ACTH Measurements

Plasma Steroid concentrations were determined using liquid chromatography-tandem mass spectrometry (LC-MS/MS) [30]. The ACTH concentration was measured using the ACTH (Mouse/Rat) ELISA kit (Abnova, Taipei, Taiwan). All measurements were carried out according to the manufacturer’s protocol. 

### 4.8. 24-h Urine Collection and GC-MS Determination of Urinary Steroids

The urine of 20-week-old mice was collected using metabolic cages (Techniplast Deutschland GmbH, Hohenpeißenberg, Germany) over 24 h. The mice were given free access to standard mouse chow and autoclaved water during the entire period of urine collection. Post collection, the urine was kept frozen at −80 °C and later used for urine metabolite analysis using gas chromatography mass spectrometry GC-MS [31]. 

### 4.9. Blood Pressure Measurements

Blood pressure was measured in eight and 20-week-old mice using a non-invasive tail-cuff method using the NIBP instrument connected to a PowerLab 4/16 system (ADInstruments, Oxford, UK). During the experiment, the animals were under the influence of light isoflurane (0.5%) and subsequently transferred to a heating plate fitted with a inhalation mask to ensure the constant inhalation of isoflurane during measurement [32]. The heartrate and mean systolic, diastolic, and arterial blood pressure values were measured three consecutive times. Data were processed and analysed using the Lab Chart Program (ADInstruments, Oxford, UK).

### 4.10. Assessment of Estrous Stage 

The estrous cycle phase was assessed from vaginal epithelial cell smears taken by vaginal flush inserting sterile water into the vaginal opening of the mouse. The method is non-invasive and reliable, in which the vaginal cells were flushed gently by introducing water into the vaginal orifice using a pipette [24]. The process was repeated a couple of times to obtain a good number of cells for investigation. A small amount of cell suspension was placed on a glass slide. The air-dried slides were stained with 0.1% crystal violet and covered with a slip. The slides were examined for different cell types under a light microscope (Appendix A).

### 4.11. Histological Analysis

To obtain tissue sections of different organs, eight mice of wildtype and homozygous genotypes were sacrificed (two males and two females of each genotype). After dissection, the organs were fixed in 4% paraformaldehyde (PFA) overnight. After fixation, the organs were washed in 1× phosphate buffered saline (PBS) for 10 min, transferred into fresh PBS, and stored at 4 °C. The organs were then embedded in paraffin at the Histology Facility of BIOTEC Dresden, Germany. Four µm sections of paraffin-embedded mouse organs were placed on glass slides and stained with hematoxylin and eosin (H&E). The microscopic images were captured using Keyence BZ-X700 (Keyence Corporation of America, Itasca, IL, USA).

### 4.12. Immunohistochemical Analysis

Immunohistochemically, the mice adrenals were analyzed using 4 µM sections obtained from cryo-frozen tissues. Adrenal glands were fixed in 4% formaldehyde for 4 h and transferred to 30% sucrose at 4 °C overnight. The following day, the organs were embedded in Tissue Tek^®^ O.C.T.TM Compound Medium (Sakura Finetek Germany GmbH, Umkirch, Germany ) and immediately frozen at −80 °C. Four µM cryosections were cut using Cryostar NX70 cryostat. Next, tissue sections were blocked with a mixture of 5% normal goat serum, 0.1% bovine serum albumin, and 0.3% Triton X in PBS for 1 h. Post blocking, sections were incubated overnight at 4 °C with primary antibody tyrosine hydroxylase (TH) (Cat# AB152, Merck Millipore, Burlington, MA, USA). Subsequently, the sections were washed with PBS and incubated with fluorescence-labelled anti-rabbit Cy3 secondary antibody. Fluorescence was examined microscopically using a Zeiss Axiovert 200M inverse light microscope (Carl Zeiss Microscopy GmbH, Oberkochen, Germany) and recorded using the AxioVision SE64 Rel. 49 software (Carl Zeiss Microscopy Deutschland GmbH, Oberkochen, Germany).

### 4.13. Data Analysis

Figures and tables were generated using GraphPad Prism version 9.3.2 for Windows (GraphPad Software, La Jolla, CA, USA), Microsoft Excel, PowerPoint, and ImageJ (NIH) (https://imagej.net/ij/).

### 4.14. Statistics

Data are presented as box plots displaying the median, first and third quartile, interquartile range, minimal and maximal values, and outliers. Differences between the wildtype (WT) and homozygous mutant (HOM) mice were analyzed using GraphPad Prism version 9.3.2. Steroid hormone concentrations below detection limits (BDL) were substituted with a 0.5× limit of detection (LOD) for statistical analysis. Tests for normal distribution were performed on all data sets using the Shapiro-Wilk test. Statistical significance for normally distributed data was analyzed using an unpaired t-test. The Mann-Whitney test was used for data which were not normally distributed. Significance *p* > 0.05 ns; *p* < 0.05 *; *p* < 0.01 **; *p* < 0.001 ***; *p* < 0.0001 ****.

## Figures and Tables

**Figure 1 ijms-25-05062-f001:**
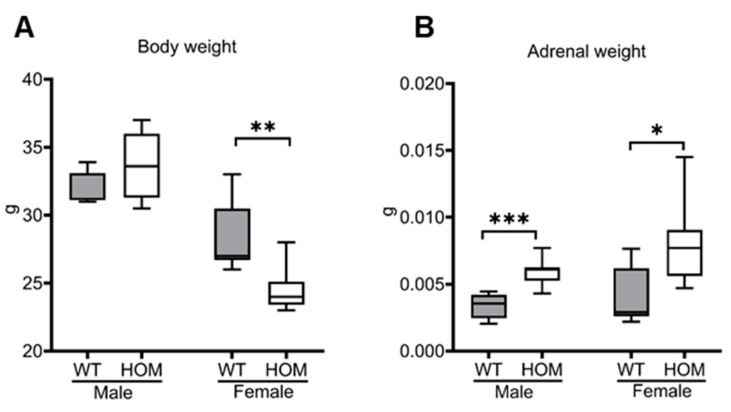
**Weights of mutant mice in comparison to its wildtype littermates at 20-weeks**. (**A**) Body weight (14 male: WT = 7, HOM = 7; 14 female: WT = 7, HOM = 7). (**B**) Adrenal weight (16 male: WT = 7, HOM = 9; 14 female: WT = 7, HOM = 7; both adrenals were weighed). All values are represented in box plots with the median as a cross bar. Statistical significance for the body weight was assessed using an unpaired t-test, while the adrenal weights were evaluated using a Mann–Whitney test. The significance levels are denoted as *p* < 0.05 *, *p* < 0.01 **, and *p* < 0.001 ***.

**Figure 2 ijms-25-05062-f002:**
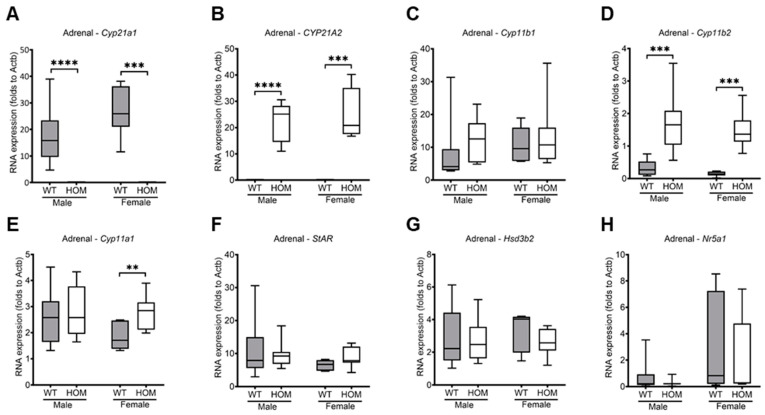
**Adrenal gene expression levels of mutant and wildtype mice at 20 weeks.** Quantitative RT-PCR was used to measure the expression levels and was performed in triplicates. (**A**) *Cyp21a1*, (**B**) *CYP21A2*, (**C**) *Cyp11b1*, (**D**) *Cyp11b2*, (**E**) *Cyp11a1*, (**F**) *StAR*, (**G**) *Hsd3b2*, and (**H**) *Nr5a1*. Relative mRNA levels are normalized to the expression of the housekeeping gene ß-Actin (*Actb*). All values are represented in box plots with the median as a cross bar. Statistical significance was determined by the Mann–Whitney test with *p* < 0.01 **, *p* < 0.001 ***, and *p* < 0.0001 ****. For analysis, the cDNA of 19 male mice (WT = 8; HOM = 11) and 15 female mice (WT = 7; HOM = 8) were used.

**Figure 3 ijms-25-05062-f003:**
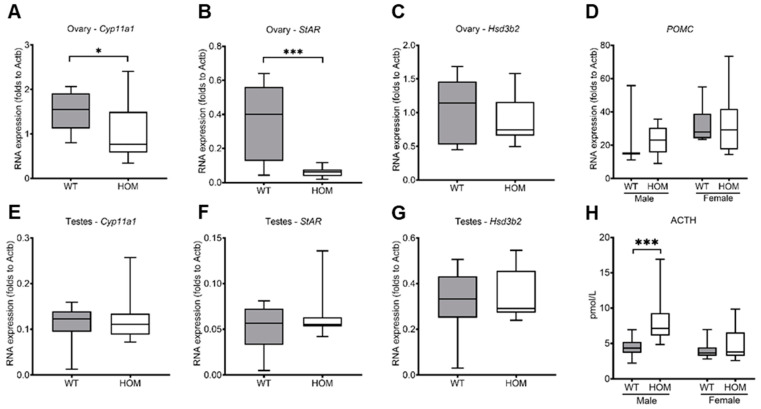
**Gonadal and pituitary gene expression levels and ACTH measurement in mutant and wildtype mice at 20-weeks.** Quantitative RT-PCR was used to measure the expression levels and was performed in triplicate. Ovaries–(**A**) *Cyp11a1*, (**B**) *StAR*, and (**C**) *Hsd3b2*, testes–(**D**) *Cyp11a1*, (**E**) *StAR*, (**F**) *Hsd3b2*, pituitary (**G**) POMC. (**H**) Plasma ACTH was measured using an ACTH (Mouse/Rat) ELISA kit. Relative mRNA levels are normalized to the expression of the housekeeping gene ß-Actin (*Actb*). All values are represented in box plots with the median as a crossbar. Statistical significance was determined using the Mann–Whitney test with *p* < 0.05 * and *p* < 0.001 ***. For analysis, the gonads of 19 male mice (WT = 8; HOM = 11) and 20 female mice (WT = 10; HOM = 10) were used. For POMC analysis, the pituitary of nine male mice (WT = 3; HOM = 6) and 14 female mice (WT = 6; HOM = 8) were used. For ACTH measurements, plasma samples of 20 male mice (WT = 11; HOM = 9) and 25 female mice (WT = 13; HOM = 12) were used.

**Figure 4 ijms-25-05062-f004:**
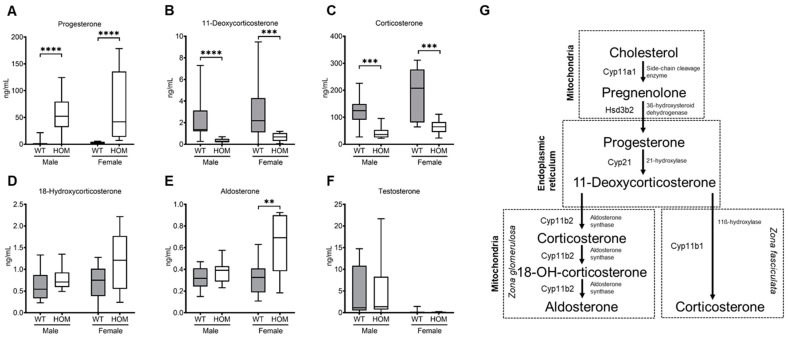
**Steroidogenic hormone concentration levels in mutant and wildtype mice at 20 weeks**. (**A**) Progesterone, (**B**) 11-Deoxycorticosterone, (**C**) Corticosterone, (**D**) 18-Hydroxycorticosterone (18-OH-corticosterone), (**E**) Aldosterone, and (**F**) Testosterone levels were measured by LC-MS/MS. (**G**) Schematic representation of the steroidogenic pathway in mice. All values are represented in box plots with the median as a crossbar. Statistical significance was determined by Mann–Whitney test with *p* < 0.01 **, *p* < 0.001 ***, *p* < 0.0001 ****. For analysis, the plasma of 23 male mice (WT = 11; HOM = 12) and 24 female mice (WT = 13; HOM = 11) were used.

**Figure 5 ijms-25-05062-f005:**
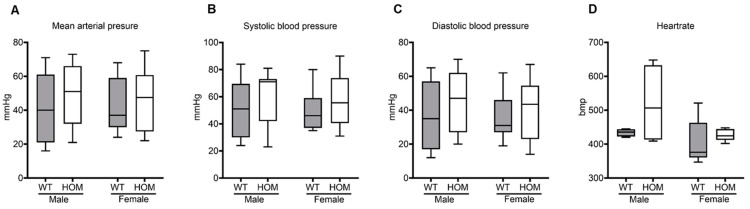
**Blood pressure measurements in mutant and wildtype mice at 20-weeks.** Blood pressure was measured three consecutive times per mouse using a non-invasive tail cuff under a constant supply of isoflurane. (**A**) Mean arterial pressure (MAP), (**B**) Systolic blood pressure (SBP), (**C**) Diastolic blood pressure (DBP), and (**D**) Heartrate. Data were processed and analysed using the Lab Chart Program. All values are represented in box plots with the median as a crossbar. Statistical significance was determined by Mann–Whitney test. For analysis 12 male mice (WT = 5; HOM = 7) and 15 female mice (WT = 7; HOM = 8) were used.

**Figure 6 ijms-25-05062-f006:**
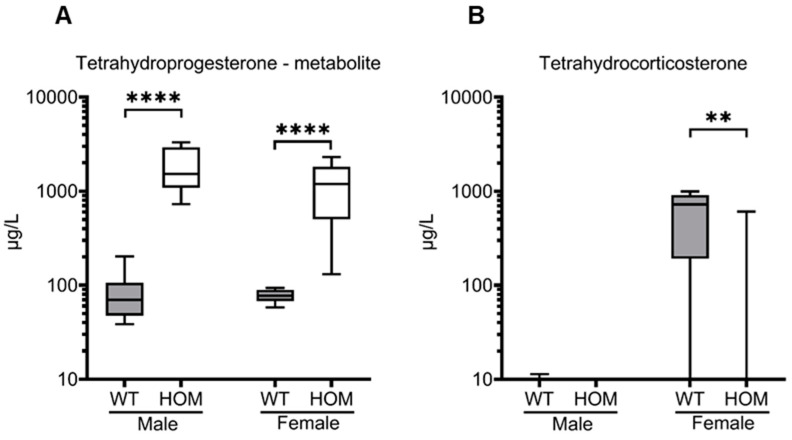
**Urine metabolite measurements in mutant and wildtype mice at 20 weeks.** Urine was collected over a 24-h period using metabolic cages and measured by gas chromatography-mass spectrometry (GC-MS). (**A**) Tetrahydroprogesterone–metabolite and (**B**) Tetrahydrocorticosterone. All values are represented in box plots with the median as a crossbar on a logarithmic scale. Statistical significance was determined by Mann–Whitney test with *p* < 0.01 ** and *p* < 0.0001 ****. For analysis, 21 male mice (WT = 10; HOM = 11) and 17 female mice (WT = 8; HOM = 9) were used.

**Figure 7 ijms-25-05062-f007:**
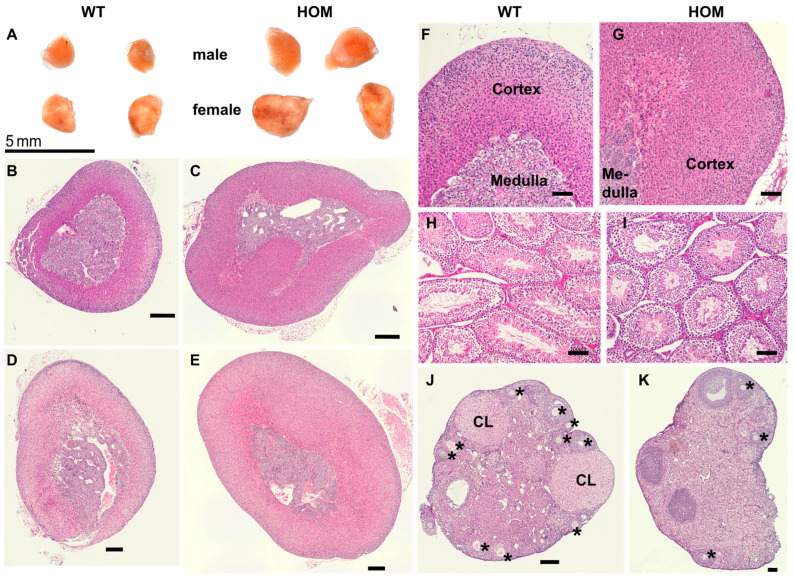
**Histological staining of organ sections of 20-week-old mice.** Macroscopic images depicting enlarged adrenals of one pair (WT, HOM) of male and female mice (**A**); Left–WT, Right–HOM. For histological sections, two animals of each (Male–WT, HOM; Female–WT, HOM) were used. Microscopic images of H&E stained sections of adrenals (**B**–**G**), testes (**H**,**I**), and ovaries with corpus luteum (CL) and follicles (*) (**J**,**K**). Images were captured in 4× (**B**–**E**,**J**,**K**), 10× (**F**,**G**), 20× (**H**,**I**). Scale bar: 300 µm (**B**,**C**), 200 µm (**D**,**E**,**J**), 100 µm (**H**,**I**,**K**).

**Figure 8 ijms-25-05062-f008:**
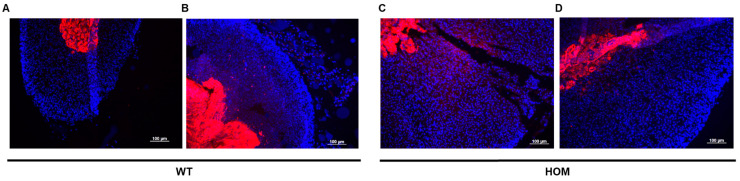
**Immunofluorescence staining of organ cryosections of 20-week-old mice.** Immunofluorescence staining of cryosections of WT (**A**,**B**) and HOM (**C**,**D**) adrenals. For specific antibody staining, sections were incubated with anti-tyrosine hydroxylase antibody and visualized by a secondary Cy3-fluorescence labelled antibody (red). No gross differences were observed for the medulla at 10×. Scale bar: 100 µm.

**Table 1 ijms-25-05062-t001:** A comparison between human CAH patients and *CYP21A2*-R484Q mouse model.

Phenotype	Human CAH Patients [10]	*CYP21A2*–R484Q Mouse Model
Adrenal hyperplasia	Present when untreated	Present
Glucocorticoid deficiency	Decreased 11-deoxycortisol and cortisol levels	Decreased levels of corticosterone and 11-deoxycorticosterone
Mineralocorticoid deficiency	Decreased levels of aldosterone in salt-wasting CAH	Not observed
Blood pressure	Hypertension in some patients	No hypertension
Accumulation of hormone precursors	Elevated levels of 17-hydroxyprogesterone	Elevated levels of progesterone
Sex-hormone imbalance	Androgens excess causing virilization	No androgen excess
Fertility issues	Fertility problems in males and females	Decreased fertility and reproductive abnormalities in female mice
Menstrual irregularities	Amenorrhea or irregular menstrual cycles	Irregular estrus cycle; females are in constant diestrus
Testicular adrenal rest tumors (TARTs)	Present in some cases	No TARTs detected in 20 weeks old animals

## Data Availability

All data generated or analyzed during this study are included in this published article or are available from the corresponding author on request.

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
