# Peer review of "A Humanized and Viable Animal Model for Congenital Adrenal Hyperplasia–CYP21A2-R484Q Mutant Mouse"

_ijms, 2024, doi:10.3390/ijms25105062_

Round 1

Reviewer 1 Report

Comments and Suggestions for Authors

In this manuscript, the authors present a novel humanized mouse model featuring the p.R484Q mutation in the CYP21A2 gene, aiming to address the lack of effective in-vivo models for pre-clinical testing of Congenital Adrenal Hyperplasia (CAH). The mutant mice exhibited several characteristic CAH phenotypes. Notably, male mutant mice retained fertility, whereas females displayed infertility due to abnormal ovarian structure. The authors suggest that their mouse model holds promise for advancing therapeutic strategies and mitigating the development of comorbidities resulting from hormone-related dysregulations and treatment-associated side effects in CAH patients.

However, certain issues have caught my attention and must be addressed.

Major concerns:

1 The authors did not conduct experiments to assess potential clinical treatments using their mouse model;

2 The authors did not evaluate comorbidities resulting from any conditional treatments.

Future experiments should be designed to address at least one of these concerns

Specific concerns:

1 Line 104: The breeding strategy involved mating mutation-carrying females with wild-type (WT) males. Why not using the humanized CYP21A2 males?

2 Line 111: The authors should conduct a Chi-square test to assess whether the distribution of different genotypes adheres to Mendelian laws.

3 Lines 132 and 160: Steridogenic should be steroidogenic

4 Lines 201 and 203: please revise this ‘Error! Reference source not found’.

5 Figure 7A: how many samples are used here? The authors should measure the weigh/volume of the adrenals and perform statistical analysis to support their claim.

6 The authors should include a table summarizing the phenotypes observed in human CAH patients alongside those observed in their mouse model. What are the commonalities and differences?

7 Statistics: The authors should provide a rationale for why they opted to use the Mann-Whitney test instead of the t-test, especially considering that the sample size for most groups exceeds six.

Author Response

Major concern 1: The authors did not conduct experiments to assess potential clinical treatments using their mouse model;

Answer: Thank you for your comment. As stated in the discussion section, our paper depicts the general characterization of these distinguished group of CAH mutant mice. The assessment of the effect of potential clinical treatments is planned for the follow-up project which is currently in an early phase.

Major concern 2:  The authors did not evaluate comorbidities resulting from any conditional treatments.

Answer: Thank you for your suggestion. The comorbidities resulting from treatments will be evaluated after subjecting the mutant mice to any treatments. The transgenic mice will be treated as a next step according to a new animal test application protocol. We can expect the results at the earliest by the end of 2025. This publication is focused on the generation and general characterization of the mouse model. Therefore, the results of potential treatments will be published in a future paper.

Specific concern 1:  Line 104: The breeding strategy involved mating mutation-carrying females with wild-type (WT) males. Why not using the humanized CYP21A2 males?

Answer: Thank you for this question. After generating the humanized CYP21A2 mouse strain, we immediately focused on the generation of the mutant mice to make these mice available as soon as possible. Our breeding strategy was based on three aspects: First, we mated the mutant mice with the non-humanized Cyp21a1 WT strain C57BL/6NCrl for the process of backcrossing to the Bl6 background. The backcross was important for us to rule out any off-target effects of the CRISPR/CAS-technology. Second, with our designed multiplex PCR we were able to easily distinguish between the non-humanized Cyp21a1 WT, heterozygous CYP21A2-R484Q and homozygous CYP21A2-R484Q. If we had used the humanized CYP21A2 males it would have been technically extremely difficult to detect the hetero- and homozygous mice. Third, the phenotypic differences between the Cyp21a1 and humanized CYP21A2 mice were fortunately only minor, so that we felt that the advantages of our breeding strategy have outreached the option with the humanized CYP212A2 mouse strain.

Specific concern 2: Line 111: The authors should conduct a Chi-square test to assess whether the distribution of different genotypes adheres to Mendelian laws.

Answer: Thank you for your suggestion. We have performed a Chi-square test, and the results are mentioned in line 117-122 and further explained it in the discussion.

Specific concern 3: Lines 132 and 160: Steridogenic should be steroidogenic

Answer: Thank you. We have made this change in lines 140 and 155.

Specific concern 4: Lines 201 and 203: please revise this ‘Error! Reference source not found’.

Answer: Thank you. We have made the changes in lines 211 and 213.

Specific concern 5: Figure 7A: how many samples are used here? The authors should measure the weigh/volume of the adrenals and perform statistical analysis to support their claim.

Answer: For histological sections, two animals of each (WT, HOM, male, female) were used as described in the methods section. From this group, a pair of adrenals of male and female animals were randomly selected for the macroscopic pictures. We included this information now also in the figure legend. Thank you for the advice. Furthermore, we measured adrenal weights and summarized the results in Figure 1b. The adrenal weights differ significantly between wildtype and homozygous animals (n≥7 animals in each group).

Specific concern 6: The authors should include a table summarizing the phenotypes observed in human CAH patients alongside those observed in their mouse model. What are the commonalities and differences?

Answer: Thank you for your suggestion. We also think this will be a valuable addition to our manuscript. The table has been added as Table 1 in the discussion section (Line 276).

Specific concern 7: Statistics - The authors should provide a rationale for why they opted to use the Mann-Whitney test instead of the t-test, especially considering that the sample size for most groups exceeds six.

Answer: Thank you for your comment.  We tested all our data for normal distribution using the Shapiro-Wilk test. We performed a paired t-test on normally distributed datasets. On not normally distributed data we used the non-parametric Mann-Whitney test. The Mann-Whitney test is a rank-based test that is less affected by outliers and extreme values making it more suitable for our study.

Reviewer 2 Report

Comments and Suggestions for Authors

In this manuscript, the authors established a humanized mouse model for congenital adrenal hyperplasia which contains the clinic relevant point mutation R484Q and they found the mutant mice showed hyperplastic adrenals. This in-vivo model has great potential to improve the therapeutic strategy for CAH patients. Although the topic of the manuscript is quite interesting and organization is quite clear, there is still some comments need to be addressed.

The authors found this mouse model carrying CYP21A R484Q mutant has the similar phenotypes with the CAH patients. And to demonstrate this model could be applicable for testing new drugs for CAH patients treatments, the authors need to provide more evidence that this model could recapitulate the phenotypes by using the current treatment method such as hormone replacement medications. It would be great to have such evidence because there are lots of differences between human and mouse in metabolism.

Author Response

Comment:  The authors found this mouse model carrying CYP21A R484Q mutant has the similar phenotypes with the CAH patients. And to demonstrate this model could be applicable for testing new drugs for CAH patients treatments, the authors need to provide more evidence that this model could recapitulate the phenotypes by using the current treatment method such as hormone replacement medications. It would be great to have such evidence because there are lots of differences between human and mouse in metabolism.

Answer: Thank you for your comment. As stated in the discussion section, our paper intended the general phenotypic characterization of these worldwide first knock-in CAH mice which stay alive without treatment. In a next project we aim to assess the effects of the classic corticoid treatment in comparison with other treatments on the mice phenotype. This project has just started and will be ongoing until the end of next year. We will certainly share these results with the research community as soon as they are available. To wait for these treatment results at this time point, would however result in an unnecessary and marked delay in sharing the information about the existence of our mutant CAH mouse model.

Reviewer 3 Report

Comments and Suggestions for Authors

In the manuscript by Thirumalasetty et al., the authors constructed a mice model of Congenital Adrenal Hyperplasia (CAH) and comprehensively analyzed the physiological indices in the mutant and wild type. The mice model of CAH could serve as a useful tools for the study of CAH pathogenesis. However, there is still certain aspects need to improve to make this study more significant.

Major commons:

1.     In Figure 2A, it’s the RNA level quantification. How’s about the protein level between mutant and wild type mice?

2.     In line 155-157, if the RNA level of POMC wasn’t significantly different between mutant animals and their wildtype littermate, how does the protein level of POMC distribute between WT and  HOM? And also, if the ACTH didn’t accurately measure because of technical issues and storage of samples post plasma collection, recollection of sample needs to be performed to quantify the ACTH.

Minor commons:

1.     In line 52, there is a redundant blank after word “severity”.

2.     In line 201 and 203, there are errors to refer to the corresponding figures.

3.     In supplementary figure 3, the sex of samples is miss in all the panels.  

4.     In Figure 7B-E, 7J-K, 8A-D, the scale bars are too small while the scale bars in Figure 7F-G is missing.

Comments on the Quality of English Language The English of the manuscript is good.

Author Response

Major concerns: 

Question 1: In Figure 2A, it’s the RNA level quantification. How’s about the protein level between mutant and wild type mice?

Answer: Thank you for this question. We conducted a Western blot analysis using adrenals from both a WT and a HOM mouse. The blot was developed using anti mouse-21-hydroxylase antibody (PA5-119068). As expected, we successfully detected a band at ~ 52 kDa in the lysate of WT mouse adrenal, whereas this band was absent in the HOM mice. Vice verse we could detect the band at ~ 52 kDa in lysate of HOM adrenal and not in lysate of WT adrenal using anti human CYP21A2 polyclonal antibody (HPA053371). This supports the measured gene expression levels of mCyp21a1 and hCYP21A2 in WT and HOM mice adrenals.

Question 2: In line 155-157, if the RNA level of POMC wasn’t significantly different between mutant animals and their wildtype littermate, how does the protein level of POMC distribute between WT and HOM? And also, if the ACTH didn’t accurately measure because of technical issues and storage of samples post plasma collection, recollection of sample needs to be performed to quantify the ACTH.

Answer: Thank you for this questions and comments. In this study, all pituitary glands were used for RNA measurements of POMC. We did our best for the appropriate storage and transportation of samples for ACTH measurements. However, using the ELISA method for assessment of ACTH levels we could successfully deduce a significant difference between the wildtype and homozygous animals in the males. For the females we did not see any changes between the WT and HOM. This data is now added in Figure 3 (H) and discussed (line 307-314). Nevertheless, we are certain that higher ACTH plasma concentrations in the long term must have caused the marked hyperplasia and hypertrophy of steroid-producing cells in the adrenal cortex of the homozygous mutant animals. This adrenal hyperplasia could be seen macroscopically and microscopically (Figure 7) and could be confirmed by the highly significant differences in adrenal weights between WT and HOM in 20-week-old animals (Figure 1).

Minor concerns: 

Concern 1: In line 52, there is a redundant blank after word “severity”.

Answer: Thank you. We removed the whitespace.

Concern 2: In line 201 and 203, there are errors to refer to the corresponding figures.

Answer: Thank you. We corrected this.

Concern 3: In supplementary figure 3, the sex of samples is miss in all the panels.

Answer: Thank you. We added the sexes in the legend of graphs of Figure S3.

Concern 4: In Figure 7B-E, 7J-K, 8A-D, the scale bars are too small while the scale bars in Figure 7F-G is missing.

Answer: Thank you for this comment and apologies for overseeing it. We have made the changes accordingly.

Reviewer 4 Report

Comments and Suggestions for Authors

I have read with great interest the manuscript entitled “A humanized and viable animal model for Congenital Adrenal Hyperplasia - CYP21A2-R484Q mutant mouse”. In this manuscript the authors present novel animal model of CAH. The hormonal status and some of the phenotype characteristics are similar to humans.  The manuscript is well-written and the data is nicely presented with multiple phenotype comparisons between the wild types and the mutant mice.

Comments –

Please add to your discussion a table summarizing the hormonal and clinical similarities and differences between human CAH and your mutant mice – for both females and males. Such a table will enable readers to clearly compare the two.

Author Response

Comment 1: Please add to your discussion a table summarizing the hormonal and clinical similarities and differences between human CAH and your mutant mice – for both females and males. Such a table will enable readers to clearly compare the two.

Answer: Thank you for your suggestion. We also think this will be a valuable addition to our manuscript. The table has been added as Table 1 (line 276).

Round 2

Reviewer 1 Report

Comments and Suggestions for Authors

No further comment

Reviewer 2 Report

Comments and Suggestions for Authors

The manuscript is now acceptable.

Reviewer 3 Report

Comments and Suggestions for Authors

The authors have addressed all my concerns. It can be published as it.